**DOI: 10.1038/ncomms12778**　　**OPEN**

# Real-time observation of DNA recognition and rejection by the RNA-guided endonuclease Cas9

Digvijay Singh[1,†], Samuel H. Sternberg[2], Jingyi Fei[3,4,†], Jennifer A. Doudna[2,5,6,7,8] & Taekjip Ha[1,3,4,†]

Binding specificity of Cas9–guide RNA complexes to DNA is important for genome-engineering applications; however, how mismatches influence target recognition/rejection kinetics is not well understood. Here we used single-molecule FRET to probe real-time interactions between Cas9–RNA and DNA targets. The bimolecular association rate is only weakly dependent on sequence; however, the dissociation rate greatly increases from $<0.006\,\mathrm{s}^{-1}$ to $>2\,\mathrm{s}^{-1}$ upon introduction of mismatches proximal to protospacer-adjacent motif (PAM), demonstrating that mismatches encountered early during heteroduplex formation induce rapid rejection of off-target DNA. In contrast, PAM-distal mismatches up to 11 base pairs in length, which prevent DNA cleavage, still allow formation of a stable complex (dissociation rate $<0.006\,\mathrm{s}^{-1}$), suggesting that extremely slow rejection could sequester Cas9–RNA, increasing the Cas9 expression level necessary for genome-editing, thereby aggravating off-target effects. We also observed at least two different bound FRET states that may represent distinct steps in target search and proofreading.

[1] Center for Biophysics and Quantitative Biology, University of Illinois at Urbana-Champaign, Urbana, Illinois 61801, USA. [2] Department of Chemistry, University of California, Berkeley, California 94720, USA. [3] Department of Physics and Center for the Physics of Living Cells, University of Illinois at Urbana-Champaign, Urbana, Illinois 61801, USA. [4] Howard Hughes Medical Institute, Baltimore, Maryland 21205, USA. [5] Department of Molecular and Cell Biology, University of California, Berkeley, California 94720, USA. [6] Howard Hughes Medical Institute, Berkeley, California 94720, USA. [7] Physical Biosciences Division, Lawrence Berkeley National Laboratory, Berkeley, California 94720, USA. [8] Innovative Genomics Initiative, University of California, Berkeley, California 94720, USA. † Present address: Department of Biophysics and Biophysical Chemistry, Johns Hopkins University School of Medicine, Baltimore, Maryland 21205, USA; Department of Biophysics, Johns Hopkins University, Baltimore, Maryland 21205, USA; Department of Biomedical Engineering, Johns Hopkins University, Baltimore, Maryland 21205, USA (D.S. and T.H.); Department of Biochemistry and Molecular Biology, University of Chicago, Chicago, Illinois 60637, USA (J.F.). Correspondence and requests for materials should be addressed to J.A.D. (email: doudna@berkeley.edu) or to T.H. (email: tjha@jhu.edu).

CRISPR (clustered regularly interspaced short palindromic repeats)–Cas systems provide adaptive immunity against foreign genetic elements in bacteria and archaea[1]. In type II systems, the Cas9 endonuclease functions together with a dual-guide RNA comprising CRISPR RNA (crRNA) and trans-activating crRNA (tracrRNA) to target 20 base pair (bp) DNA sequences (cognate sequence) for double-stranded cleavage[2]. Efficient targeting requires RNA–DNA complementarity as well as a specific motif flanking the target sequence called the PAM (protospacer adjacent motif, 5′-NGG-3′ for Streptococcus pyogenes Cas9)[2–4]. Cas9–RNA complexes have proven to be extremely versatile tools for genome-engineering applications[5], and minimizing off-target effects[6,7] remains an active area of study.

Numerous studies have assessed off-target DNA binding and cleavage by the Cas9–RNA complex, both in vitro and in vivo[2,4,8–43]. While subtly different conclusions have been reached depending on the exact method of analysis, these studies agreed about specificity being heavily influenced by the presence of a PAM, a 7–12 bp-long seed sequence proximal to the PAM, and the concentration of Cas9 and guide-RNA. Most of previous studies lacked dynamic information on DNA targeting; yet, in order to improve the efficacy of processing only the correct targets, we need such information on targeting dynamics. Single-molecule methods are ideal for this task because they can detect wide-ranging interactions (transient to long-lived) and identify multiple states in real time[44]. Moreover, they can be used to obtain the dynamic information on short and specific DNA sequences, thus enabling the sequence-specific estimation of various kinetic parameters[45–47]. Several single-molecule studies have examined sequence specificity in CRISPR targeting[40–43,48,49]. Here we report a systematic investigation of the binding and dissociation kinetics of Cas9–RNA as a function of sequence mismatches to determine how quickly the cognate sequence is recognized and how quickly partially matching sequences are rejected.

## Results

**DNA interrogation by Cas9–RNA.** We used single-molecule fluorescence resonance energy transfer[50,51] (smFRET) to directly observe individual Cas9–RNA complexes binding to DNA targets in real time. Donor (Cy3) and acceptor (Cy5) fluorophores were conjugated to modified nucleotides in the DNA target and crRNA, respectively, so that FRET between them would report on Cas9–RNA binding to the DNA (Fig. 1a and Supplementary Fig. 1). Fluorescence labelling did not compromise target cleavage (Supplementary Fig. 2). After introducing 20 nM Cas9–RNA complexes to cognate DNA target molecules immobilized on passivated microscope slides, two distinct populations were observed centred at FRET = 0.92 and 0, respectively (Fig. 1b,c). The labelling sites are separated by 30 Å (ref. 52; Supplementary Fig. 1), consistent with the observation of the high FRET value upon Cas9–RNA binding. In control experiments using a non-cognate (fully mismatched) DNA target with PAM (Supplementary Table 1), or guide-RNA without Cas9, the 0.92 FRET state was not observed (Fig. 1c). Therefore, we assigned the 0.92 FRET state to a stably formed Cas9–RNA–DNA complex. The high FRET state was long-lived, with a lifetime (>3 min) limited only by fluorophore photobleaching (Supplementary Fig. 3d). A catalytically dead Cas9 mutant (dCas9; D10A and H840A mutations[2,3]) showed signal indistinguishable from active Cas9 (Supplementary Fig. 3), indicating that DNA products remain tightly bound after cleavage as was observed previously[4] (Supplementary Fig. 2). To capture the moment of binding, we added Cas9–RNA into the flow cell during data acquisition. FRET

efficiency increased from 0 to 0.92 in a single step (Fig. 1d), suggesting that any intermediates on-path to target binding, if present, cannot be resolved at the time resolution of our measurements (0.1 s).

**Effect of DNA target mismatches on Cas9–RNA binding.** We next examined how DNA targets with imperfect RNA–DNA complementarity are discriminated against and rejected by Cas9–RNA. We prepared a series of donor-labelled, fully duplexed DNA containing mismatches relative to the guide-RNA (Supplementary Table 1 and Fig. 2a). The mismatches were introduced either from the PAM-proximal side or from the PAM-distal side, and are denoted using the naming convention $x–y_{mm}$ where $x$th through $y$th bps are mismatched. The fraction of DNA bound by Cas9–RNA (ratio between counts with FRET > 0.75 and total counts in FRET histograms) remained identical to the cognate DNA up to 12 PAM-distal mismatches ($17–20_{mm}$, $13–20_{mm}$, $12–20_{mm}$, $11–20_{mm}$, $10–20_{mm}$, $9–20_{mm}$; Fig. 2b,d). The bound state remained stable, with the observed lifetimes limited only by fluorophore photobleaching (Supplementary Fig. 3d). A large decrease in the bound fraction occurred only when the number of mismatches from the distal end exceeded 13 bp ($7–20_{mm}$, $6–20_{mm}$, $5–20_{mm}$), corresponding to less than seven matched bp from the PAM-proximal end. In contrast, even 2 bp mismatches from the PAM-proximal end ($1–2_{mm}$) were deleterious for Cas9–RNA binding and binding to 4 bp PAM-proximal mismatches ($1–4_{mm}$) was indistinguishable from binding to fully mismatched ($1–20_{mm}$), underscoring the importance of the PAM-proximal seed region (Fig. 2c,d).

**Different DNA-binding modes of Cas9–RNA.** For DNA targets to which Cas9–RNA binds weakly, we observed a second bound state with a mid-FRET peak at ∼0.42, in addition to the 0.92 high FRET state. Single-molecule time trajectories (Fig. 3a) and transition density plots reporting on the relative transition frequencies after hidden Markov modelling analysis[53] (Fig. 3b and Supplementary Figs 4–6) revealed reversible transitions between the unbound (FRET < 0.2) state and both mid and high FRET = 2 bound states, and lifetime analysis as a function of Cas9–RNA concentration confirmed that transitions are due to Cas9–RNA association/dissociation events (Supplementary Figs 7 and 8). The mid FRET was more frequently observed as the number of mismatches increased (Fig. 3b and Supplementary Fig. 4), and were also observed for DNA targets without PAM or without matching sequence, indicating that it does not require either (Supplementary Figs 9 and 10). The high FRET state was rarely observed without PAM (Supplementary Fig. 10). We propose that the Cas9–RNA has two binding modes (Fig. 4). The mid-FRET state (sampling mode) likely does not involve RNA–DNA heteroduplex formation and may represent a mode of PAM surveillance. It is possible that local diffusion may give rise to the time-averaged FRET value of the mid-FRET state. Sequence-independent sampling of DNA target for PAM can occur at multiple locations in the DNA target, and the mid-FRET state, representative of this sampling, expectedly had a broad FRET distribution (Supplementary Figs 4–6). If PAM is recognized during transient biding in the sampling mode, RNA–DNA heteroduplex formation follows, resulting in the high FRET state. Multimodal binding kinetics were also observed for Cascade in Type I CRISPR systems; however, its shorter-lived binding mode plays a priming function, which is absent for Cas9 (ref. 42).

**Effect of mismatches on binding and dissociation kinetics.** Survival probability distributions of dwell times in the bound

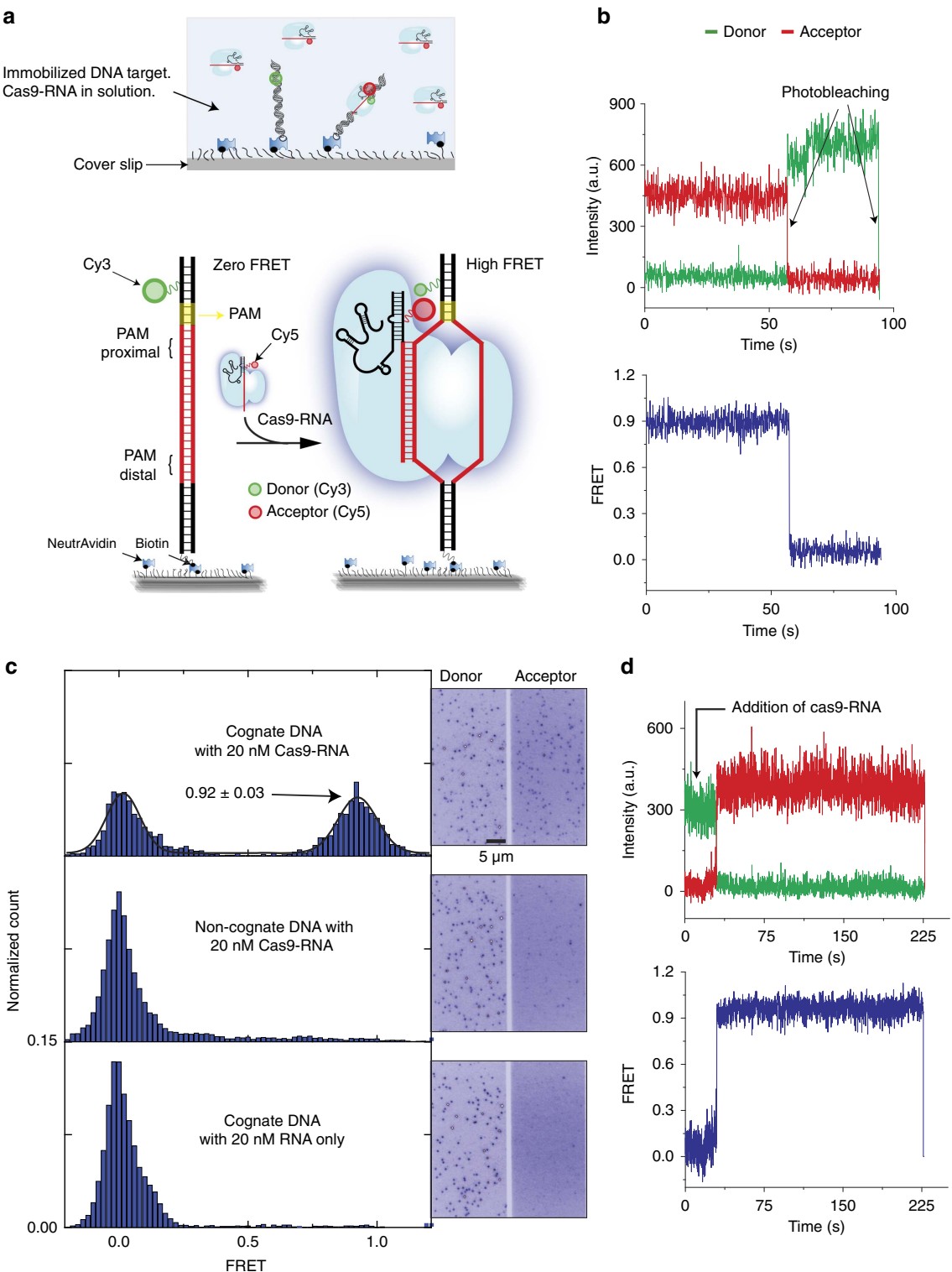

**Figure 1 | Cas9–RNA binding to a cognate sequence. (a)** Schematic of single-molecule FRET assay. High-FRET signal resulted when Cas9 in complex with an acceptor (Cy5)-labelled guide-RNA (Cas9–RNA) bound a surface-immobilized, donor (Cy3)-labelled target DNA that contains the cognate sequence (red DNA segment) and PAM (yellow segment). **(b)** A representative smFRET time trajectory of a stably bound Cas9–RNA in the presence of 20 nM Cas9–RNA in solution. **(c)** FRET histograms obtained with cognate DNA (top) and negative controls with a non-cognate DNA (middle) and with RNA only (without Cas9; bottom). The number of molecules included ranged from 568 to 1,314. Corresponding images of donor and acceptor channels are shown. **(d)** A representative smFRET time trajectory of real-time binding of Cas9–RNA in a single step after 20 nM Cas9–RNA is added at the time point indicated.

state (FRET > 0.2) before transitioning to the unbound state were best described by a double-exponential decay (Supplementary Fig. 11). The amplitude-weighted lifetime of the bound state ($\tau_{avg}$) decreased precipitously even with just 2 bp PAM-proximal

mismatches, likely because the R-loop failed to extend beyond the mismatches. In contrast, 12 bp mismatches were necessary from the distal end for any detectable decrease in $\tau_{avg}$ (Fig. 3c). Similar pattern was also observed for lifetime of transitions from high to

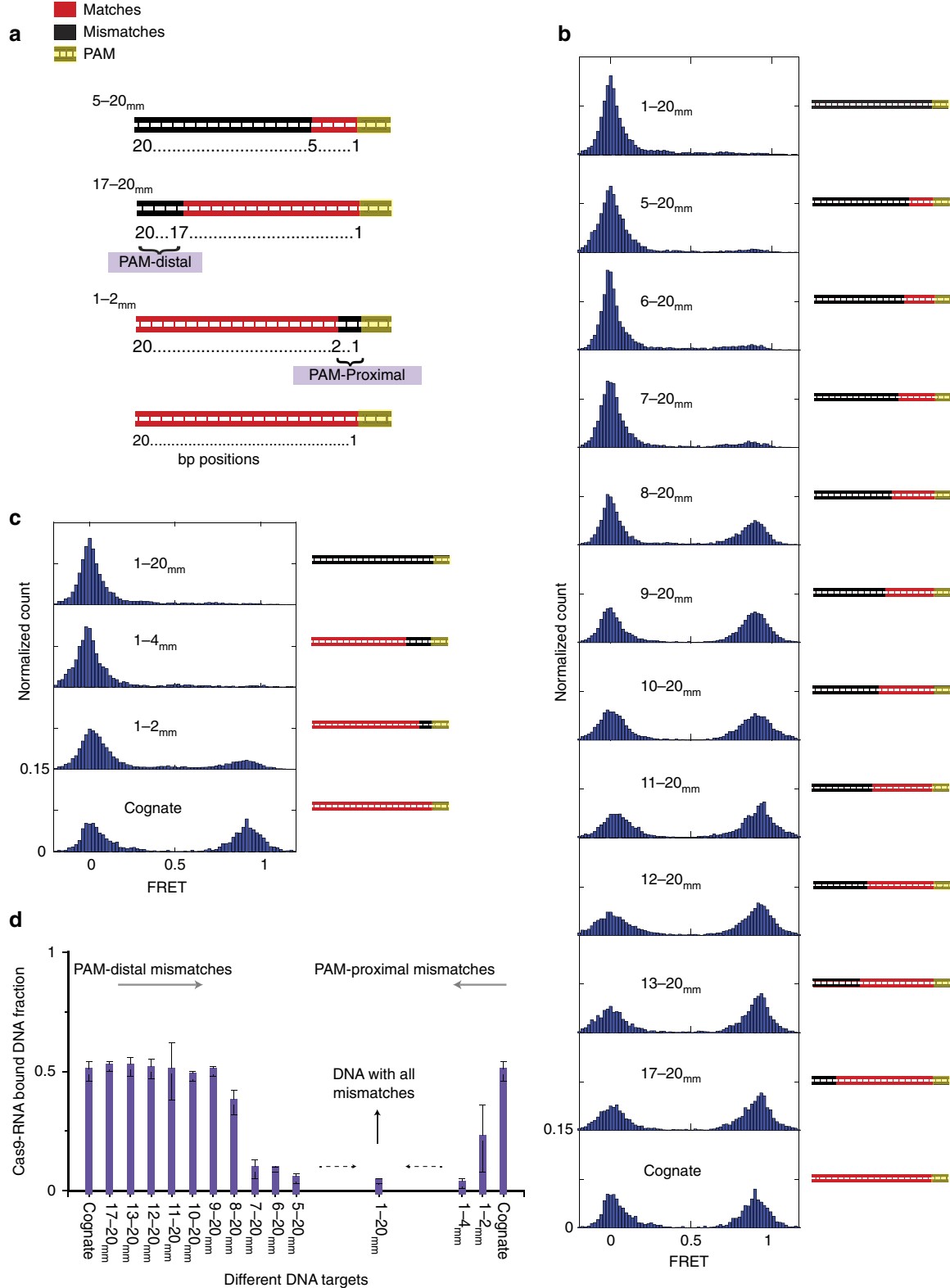

**Figure 2 | Cas9–RNA binding to DNA with proximal or distal mismatches. (a)** A series of fully duplexed DNA targets with a varying number of mismatches (black segments) relative to the guide RNA. An $x$–$y_{mm}$ target has a contiguous mismatch running from position $x$ to $y$ relative to PAM. **(b,c)** FRET histograms of Cas9–RNA binding to DNA constructs carrying PAM-distal **(b)** and PAM-proximal **(c)** mismatches. The number of molecules for each histogram ranged from 568 to 3,053. [Cas9–RNA] = 20 nM. **(d)** The fraction of Cas9–RNA-bound DNA molecules for different DNA targets. All the data shown in the figure are from independent experiments and error bars represent s.d. for $n = 3$ ($n = 2$ for few sets).

zero FRET state, whereas the lifetimes of the mid to zero FRET state remained short for all DNA targets tested, on average ~0.1 s (Fig. 3c), supporting our proposal that the mid-FRET state is a

sampling mode that does not require sequence recognition. In contrast to the bound state lifetimes, the lifetimes of the unbound state were only weakly dependent on sequence (Fig. 3c and

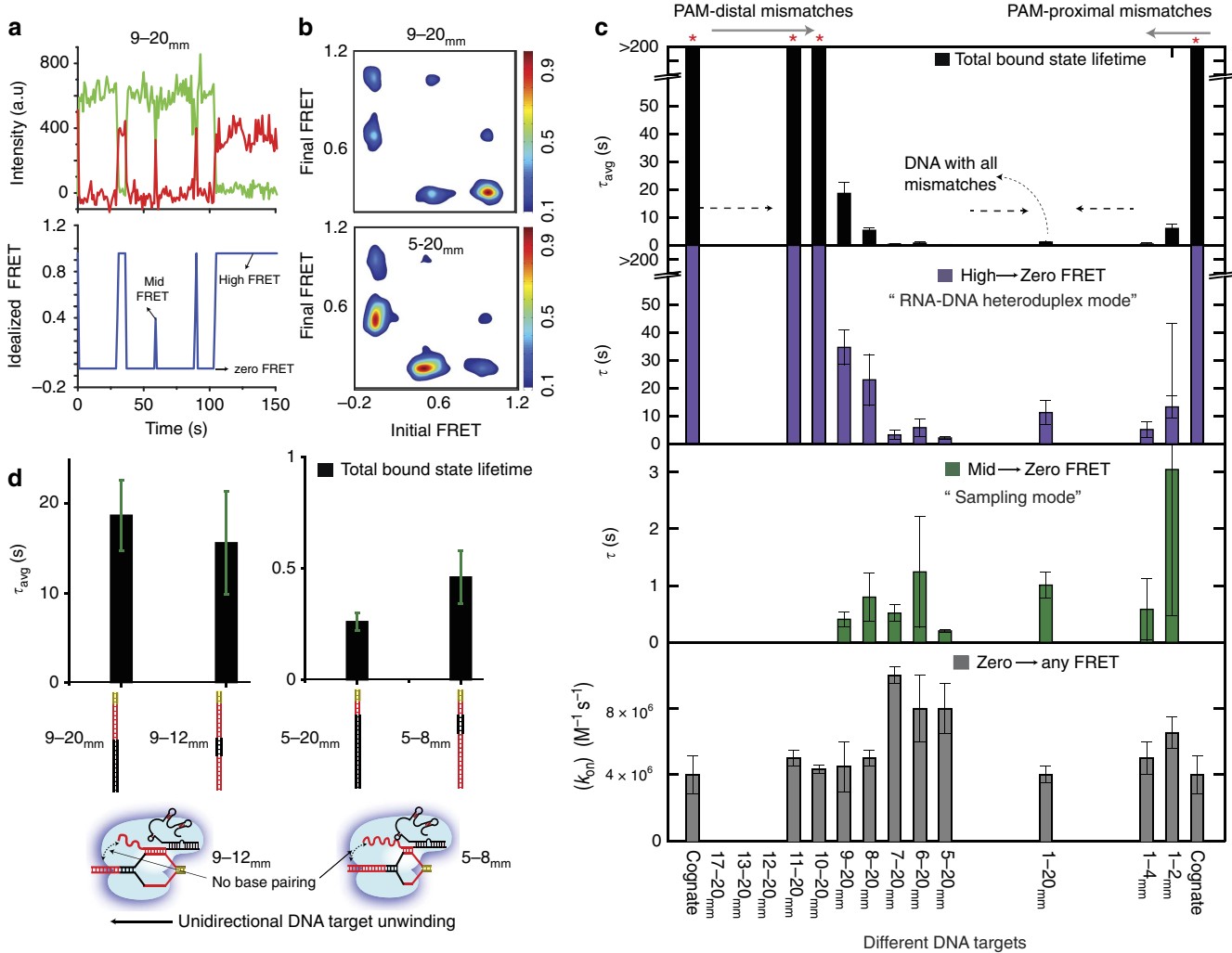

**Figure 3 | Cas9–RNA-bound state lifetimes for different DNA targets.** (**a**) smFRET time trajectory (donor and acceptor intensities, top, and idealized FRET via hidden Markov modelling (HMM) analysis, bottom) for 9–20$_{mm}$ DNA target in the presence of 20 nM Cas9–RNA. Reversible Cas9–RNA association to high- and mid-FRET states and disassociation to zero-FRET state are shown. (**b**) Transition density plots show relative transition frequencies between different FRET states for 9–20$_{mm}$ and 5–20$_{mm}$ DNA targets. [Cas9–RNA] = 20 nM. (**c**) The amplitude-weighted lifetime, $\tau_{avg}$, of the putative bound state, lifetime of high to zero and mid to zero FRET state transitions and biomolecular rate association constants for different DNA targets. On the basis of our model, the mid- and high-FRET states correspond to sampling and RNA–DNA heteroduplex modes, respectively. (**d**) Lifetime comparison of DNA targets with the respective DNA targets containing mismatches after the roadblock. All the data shown in the figure are from independent experiments and error bars represent s.d. for $n = 3$ ($n = 2$ for few sets).

Supplementary Fig. 9), yielding the bimolecular association rate constant ($k_{on}$) of $\sim 6 \times 10^6 \, M^{-1} \, s^{-1}$ with some reduction for DNA targets without PAM. Overall, our kinetic analysis showed that mismatches affect Cas9–RNA binding mainly through changes in the dissociation rate. Complete kinetic model along with rates of transitions between different states for some DNA targets are in Fig. 4 and Supplementary Fig. 12.

The relative importance of PAM-proximal bps over the PAM-distal bps Fig. 2d,3c supports the model of unidirectional extension of the RNA–DNA heteroduplex starting from the PAM-proximal end[4,48]. For 5–20$_{mm}$ that has a maximum of 4 bp heteroduplex extension from PAM, the bound state lifetime was $\sim 0.5$ s, that is, such potential targets are rapidly rejected. The bound state lifetime increased to $\sim 8$ s for 8–20$_{mm}$ with 7 bp heteroduplex, and to $\sim 16$ s for 9–20$_{mm}$ with 8 bp heteroduplex. For 9 bp or more heteroduplexes, the measured lifetime was limited by photobleaching lifetime of $\sim 3$ min (Supplementary Fig. 3). Therefore, DNA sequences with nine or more matching

bp from the PAM-proximal end have extremely long times, and Cas9–RNA would be unable to reject such sequences rapidly. A prediction of this model is that inserting a roadblock of mismatches near this boundary would prematurely terminate heteroduplex extension such that dissociation kinetics would be independent of the presence of a matched sequence beyond the block. To test this prediction, we created two 'roadblock' targets, 9–12$_{mm}$ and 5–8$_{mm}$. Indeed, the binding fraction and the lifetime of the bound state (Supplementary Fig. 13 and Fig. 3d) for 9–12$_{mm}$ and 5–8$_{mm}$ were similar to those of 9–20$_{mm}$ and 5–20$_{mm}$, respectively, confirming our prediction.

## Discussion
A previous single-molecule[48] study that investigated the Cas9—RNA-induced RNA–DNA heteroduplex formation via magnetic tweezers observed 11 PAM-proximal matches to be sufficient for stable RNA–DNA heteroduplex formation for StCas9 (the Cas9

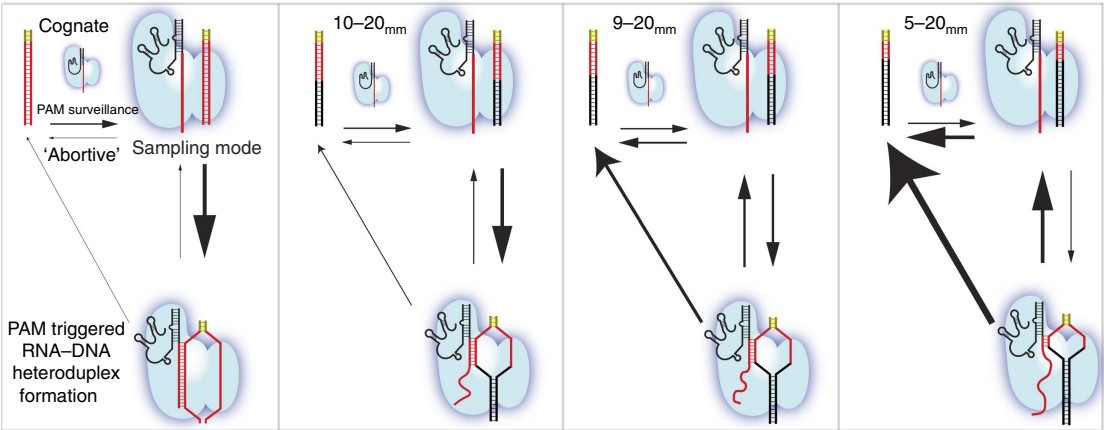

**Figure 4 | The proposed model of bimodal Cas9–RNA binding along with the kinetics of Cas9–RNA DNA targeting as a function of mismatches.** Cas9-RNA targeting occurs in predominantly two steps. The first step, that is, initial Cas9-RNA binding to DNA target is a transient PAM surveillance step, independent of the DNA sequence. In the second step, following the PAM detection, Cas9-RNA proceeds to form RNA-DNA heteroduplex in a unidirectional manner, that is, from PAM-proximal to PAM-distal end. Rate of transition between various Cas9-RNA targeting steps for different DNA targets is indicated by the size of arrows.

orthologue from *Streptococcus thermophilus*). In our current study using SpCas9 (from *S. pyogenes,* simply referred to as Cas9), we found 9–10 PAM-proximal matches to be sufficient for ultrastable Cas9–RNA binding. The stability of RNA-guided CRISPR enzymes and DNA targets depends on energetic contributions of RNA–DNA heteroduplex and interactions between the DNA target and amino-acid residues of the CRISPR enzymes. The latter has been fine-tuned[54,55] through protein engineering to create more specific Cas9 variants and the small differences between different Cas9 orthologues may stem from the variations in the interactions between the DNA target and protein residues.

A two-step mechanism of Cas9–RNA binding involving PAM surveillance in the sampling mode and RNA–DNA heteroduplex formation upon PAM recognition (Fig. 4) is also supported by structural analysis of Cas9 and Cas9–RNA–DNA ternary complexes, in which interactions between PAM-interacting amino-acid motifs in Cas9 and the PAM of the DNA target precede and guide the further RNA–DNA heteroduplex formation[19,52,56]. Our observation that the heteroduplex lifetime increases greatly between 6 and 8 bps can be explained by the recently determined Cas9–RNA structure[56], in which Watson–Crick faces of eight PAM-proximal nucleotides are solvent-exposed, thus primed for heteroduplex formation. Once an RNA–DNA heteroduplex of 8 bp or more is formed, Cas9–RNA establishes a stable complex with the DNA, regardless of PAM-distal mismatches. Therefore, Cas9–RNA is unable to rapidly reject such off-target DNA, which it cannot cleave, and is sequestered by off-target DNA, limiting the speed of genome editing. This effect would increase the minimal amount of Cas9–RNA required for genome editing, and may in turn lead to an increase in off-target cleavage. For applications requiring binding only, for example, genome decoration or gene regulation, binding specificity will be almost entirely determined by the first 8 or 9 bps away from PAM, greatly reducing the ability to target well-defined sequences in a large genome. For example, we found that 1,126 positions in the human reference genome match PAM plus 8 bp in the sequence we used (Supplementary Table 2). For future improvements in Cas9 proteins, we suggest that one should focus on rapid rejection of such off targets. Our observations may also further inform the design of the guide-RNA and the DNA targets with minimal off-target effects[57–68].

## Methods

**Preparation of DNA targets.** All DNA oligonucleotides were purchased from Integrated DNA Technologies (Coralville, IA 52241). The Cy3 label in the DNA target is located 3 bp upstream of the PAM (5′-NGG-3′) and was achieved via conjugation of Cy3 N-hydroxysuccinimido (NHS) to an amino group attached to a modified thymine through a C6 linker (amino-dT). The entire panel of DNA targets used in our measurements is available in Supplementary Table 1. A 22-nucleotide-long biotinylated adaptor strand was used for surface immobilization (Supplementary Fig. 1b). DNA targets were prepared by mixing all three component strands and heating to 90 °C followed by cooling to room temperature over 3 h.

**Expression and purification of Cas9 and dCas9.** The protein purification protocol was adapted from pervious methods[2,19] as follows: a fusion construct inserted into a custom pET-based expression vector was used for protein expression. The fusion construct consisted of the sequence encoding Cas9 (Cas9 residues 1–1,368 from *S. pyogenes*) and an N-terminal decahistidine-maltose-binding protein (His10-MBP) tag, followed by a peptide sequence containing a tobacco etch virus protease cleavage site. The fusion protein was expressed in *Escherichia coli* strain BL21 Rosetta 2 (DE3; EMD Biosciences), grown in 2xYT medium at 18 °C for 16 h following induction with 0.5 mM isopropyl β-D-1-thiogalactopyranoside (IPTG). The harvested cells were lysed in 50 mM Tris pH 7.5, 500 mM NaCl, 5% glycerol, 1 mM tris(2-carboxyethyl)phosphine (TCEP), supplemented with protease inhibitor cocktail (Roche), and then homogenized (Avestin). Following ultracentrifugation, the supernatant-clarified cell lysate was separated from the cellular debris and bound in batch to Ni-NTA agarose (Qiagen). The resin was washed extensively with 50 mM Tris pH 7.5, 500 mM NaCl, 10 mM imidazole, 5% glycerol and 1 mM TCEP, and the bound protein was eluted in a single step with 50 mM Tris pH 7.5, 500 mM NaCl, 300 mM imidazole, 5% glycerol and 1 mM TCEP. TEV protease was added to the elutant and cleavage of the protein fusion was allowed to proceed overnight. Cas9 was then dialysed into Buffer A (20 mM Tris-Cl pH 7.5, 125 mM KCl, 5% glycerol and 1 mM TCEP) for 3 h at 4 °C, before being applied on a 5 ml HiTrap SP HP sepharose column (GE Healthcare). After washing with Buffer A for three column volumes, Cas9 was eluted using a linear gradient from 0 to 100% Buffer B (20 mM Tris-Cl pH 7.5, 1 M KCl, 5% glycerol and 1 mM TCEP) over 20 column volumes. The protein was further purified using gel filtration chromatography on a Superdex 200 16/60 column (GE Healthcare) in Cas9 Storage Buffer (20 mM Tris-Cl pH 7.5, 200 mM KCl, 5% glycerol and 1 mM TCEP). Cas9 was stored at − 80 °C. Catalytically dead Cas9 (dCas9; D10A/H840A mutations) was prepared with the same protocol.

**Preparation of guide-RNA and Cas9-RNA.** The guide-RNA consists of crRNA and tracrRNA. The crRNA with an amino-dT was purchased from Integrated DNA Technologies and was labelled using Cy5-NHS. The tracrRNA was prepared using *in vitro* transcription as described previously[4]. The guide-RNA was assembled freshly for each experiment by mixing equimolar amount of Cy5-labelled crRNA with tracrRNA, heated to 80 °C followed by slow cooling to room temperature. The guide-RNA was then complexed with Cas9 (two to three times the stoichiometric amount of guide-RNA) to form the Cas9–RNA complex for use in imaging experiments. RNA sequences are available in Supplementary

Table 1. A detailed schematic of the DNA and the Cas9–RNA design can be found in the Supplementary Fig. 1. The Cas9–RNA activity on the cognate sequence used in this study was characterized previously[4]. Our biochemical assays showed that fluorophore labelling in the DNA target or crRNA had not impaired DNA target cleavage. (Supplementary Fig. 2).

**Single-molecule detection and data analysis.** Cy3-labelled DNA targets were immobilized on the polyethylene glycol-passivated surface using neutravidin–biotin interaction. The DNA target molecules were then imaged in the presence of Cy5-labelled Cas9–RNA (referred to as Cas9–RNA for brevity here) using the total internal reflection fluorescence microscopy. Imaging was performed at room temperature in a buffer (20 mM Tris-HCl, 100 mM KCl, 5 mM MgCl$_2$, 5% (v/v) glycerol, 0.2 mg ml$^{-1}$ bovine serum albumin, 1 mg ml$^{-1}$ glucose oxidase, 0.04 mg ml$^{-1}$ catalase, 0.8% dextrose and saturated Trolox ($\sim$3 mM)). The time resolution for all the experiments was 100 ms, unless stated otherwise. Detailed methods of smFRET data acquisition and analysis were described previously[51]. The FRET efficiency of a single molecule was approximated as $FRET = I_A/(I_D + I_A)$, where $I_D$ and $I_A$ are the background and leakage-corrected emission intensities of the donor and acceptor, respectively.

**FRET histograms and Cas9–RNA-bound DNA fraction.** The first five frames (100 ms each) of each of the molecule's FRET time trajectories were used as data points to construct the FRET histograms. The first 10 frames were used for the FRET histograms in Supplementary Fig. 2. The Cas9–RNA-bound DNA fraction was calculated as the fraction of data points with FRET $> 0.75$ and the total number of data points in the FRET histograms. For each DNA target, the single-molecule FRET time trajectories from independent experiments were combined together to construct the FRET histograms as described.

**Lifetime analysis of bound and unbound states.** To confirm that the FRET signal indeed reports on Cas9–RNA binding, the lifetimes of the zero FRET (FRET $< 0.2$) and the putative bound state (mid- and high-FRET states taken as a single state, FRET $> 0.2$) were determined as a function of Cas9–RNA concentration (Cas9–RNA). On the basis of this cutoff of FRET $= 0.2$, the survival probability of the zero FRET state versus time could fit well with a single exponential decay, and the decay rate increased linearly with [Cas9–RNA]. In contrast, the survival probability versus time for the bound state had to be fit with a double exponential decay and the decay rates did not depend on [Cas9–RNA] (Supplementary Fig. 2). Therefore, a bimolecular association/disassociation kinetics was used for the analysis of DNA binding by Cas9–RNA.

$$\text{DNA Target} + \text{Cas9-RNA} \underset{k_{\text{off}}}{\overset{k_{\text{on}}}{\rightleftharpoons}} \text{Cas9-RNA-DNA} \qquad (1)$$

$$k_{\text{binding}}\left(\text{s}^{-1}\right) = k_{\text{on}}\left(\text{M}^{-1}\text{s}^{-1}\right) \times [\text{Cas9-RNA}](\text{M}) \qquad (2)$$

**Lifetime of the bound state via thresholding.** In order to perform an unbiased analysis of apparently three-state FRET fluctuations observed from binding-challenged DNA targets, we employed hidden Markov model analysis and generated idealized FRET time trajectories[53], assuming that there are three distinct FRET states (high-, mid- and zero-FRET states). To estimate the lifetime of the putative bound states, the survival probability of all the bound state events (mid- and high-FRET states taken as a single state, FRET $> 0.2$) versus time was fit using a double exponential decay profile ($A_1 \exp(-t/\tau_1) + A_2 \exp(-t/\tau_2)$) (Supplementary Fig. 11). The final bound state lifetime ($\tau_{\text{avg, observed}}$) is an amplitude-weighted average of two distinct lifetimes $\tau_1$ and $\tau_2$, that is, $\tau_{\text{avg, observed}} = A_1\tau_1 + A_2\tau_2$ ($k_{\text{observed}} = 1/\tau_{\text{avg, observed}}$).

**Association rates.** We determined the observed rates of binding $k_{\text{binding}}$ using two independent methods. First, the Cas9–RNA binding events were captured in real time by flowing Cas9–RNA into the sample chamber with immobilized DNA target molecules (Supplementary Fig. 7a). Second, for the binding-challenged DNA targets that showed reversible association/disassociation, the smFRET time trajectories obtained under steady-state conditions were used to extract the unbound state duration between adjacent binding events (Supplementary Fig. 7b). These dwell times in the unbound state were then used to get the rate of association by fitting their survival probability distribution to a single exponential decay (Supplementary Fig. 8a).

**Rates of transitions between different states.** Generation of idealized FRET time trajectories using the hidden Markov model[53] yielded three different FRET states (zero, mid and high) along with the probabilities of transitions between the various FRET states. The log of transition probabilities between any two states was used to estimate the mean transition probability between the two given states, which was then used to estimate the rate as following:

$$k_{\text{A-B}}\left(\text{s}^{-1}\right) = T_{P\ (\text{A-B})} \times \text{Sampling rate of image acquisition}$$

where $k_{\text{A-B}}$ is the rate of transition from state A to B and $T_{P\ (\text{A-B})}$ is the mean probability of transition from A to B.

If each frame is acquired over 0.1 s, then the sampling rate (1/0.1) $= 10\,\text{s}^{-1}$.

**Correction factors.** Because the high FRET state was very long-lived for certain DNA targets (that is, $8\text{-}20_{\text{mm}}$, $9\text{-}20_{\text{mm}}$, $9\text{-}12_{\text{mm}}$, $1\text{-}2_{\text{mm}}$), their dwell times were not accurately captured because of photobleaching-induced truncation of smFRET time trajectories. The same is true for the dwell time of the unbound state. We made the following correction to obtain the actual rate. $k_{\text{actual}} = k_{\text{observed}} - k_{\text{photobleach (high/zero FRET state)}}$ where $k_{\text{observed}}$ is the rate calculated above and $k_{\text{photobleach (high/zero FRET state)}}$ is the rate of photobleaching of the high- or zero-FRET state. Finally, we obtain $\tau_{\text{avg}} = 1/k_{\text{actual}}$.

**Counts of DNA target sequences in human genome.** The human genome assembly (GRCh38.p6) was analysed using custom MATLAB scripts to calculate the total occurrences of DNA target sequences used in this study, which is referred to as the actual count (Supplementary Table 2). The total number of occurrences expected for a sequence, assuming a random distribution of A, T, G and C nucleotides, is referred to as the probabilistic count and is calculated as follows: probabilistic count $= (\frac{1}{4})^n \times$ total number of bp in human genome (3.2 billion) where $\frac{1}{4}$ is the probability of occurrence of any given nucleotide at a position in the sequence and $n$ is the number of bp in the genome.

**Data availability.** Any additional data that support the findings of this study are available from the corresponding author upon request.

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

## Acknowledgements

We thank current and past members of the Ha and Doudna group for various suggestions. The project was supported by grants from the National Science Foundation (PHY-1430124 to T.H. and 1244557 to J.A.D.) and National Institutes of Health (GM065367; GM112659 to T.H.); T.H. and J.A.D. are investigators with the Howard Hughes Medical Institute.

## Author contributions

D.S., S.H.S., J.F., T.H. and J.A.D. designed the experiments. D.S. conducted all the single-molecule experiments and synthesized guide-RNA. S.H.S. prepared Cas9, dCas9 and guide RNAs, and conducted biochemical DNA cleavage assays. D.S. and J.F. performed the data analysis. All authors discussed the data; D.S., S.H.S., J.F. and T.H. wrote the manuscript.

## Additional information

**Competing financial interests:** S.H.S and J.A.D. are inventors on a related patent application. The other authors declare no competing financial interests.

