## [Peer review file · Nature Communications]

Reviewers' comments:

Reviewer #1 (Remarks to the Author):

Singh et al. use single-molecule FRET measurements to investigate binding of spCas9 to DNA substrates that carry different numbers of mismatches at PAM proximal or PAM distal side of protospacer. This is high quality experimental work and manuscript is written in a clear and understandable manner. Central conclusion drawn from obtained results is that match between crRNA and PAM proximal region of only 8 nts is required to obtain long-lived Cas9 binding while for shorter matches Cas9 binding becomes suddenly short-lived. Very similar conclusion has been drawn in previous work (Szczelkun et al.) using Cas9 from *S. thermophilus*. For this Cas9 however >11 bp were required to obtain stable R-loop. Also a number of genome-wide ChipSeq experiments revealed that stable Cas9 binding is obtained despite extensive mismatches in PAM distal region. It is therefore very surprising to see that present manuscript does not discuss obtain results with respect to previous studies ranging from biochemistry to genome-wide observations.

Detailed comments:

- 1) The manuscript cites very few references despite rapidly growing number of studies that investigate Cas9 binding. As already stated above authors should compare in much more detail their achievements with respect to previous studies in CRISPR-field and to what extent the authors go beyond. The manuscript will certainly benefit from more comprehensive comparison that could also include R-loop formation in other CRISPR systems.
- 2) The authors report second bound state. How clearly defined is that state? It appears to me that authors only allow 3 states when obtaining their idealized FRET trajectories (e.g. Suppl. Fig 4b). Could it in reality be that second state actually consists of more states? Second state is very broadly distributed in transition density plots.
- 3) Please state more clearly what interpretation of precipitously decreased life time for PAM-proximal mismatches is. I suspect that in these cases R-loops that reach PAM-distal end are not forming at all, such that only target scanning occurs? Or do authors suspect destabilized R-loops due to presence of the PAM proximal mismatches?
- 4) Figure 3 is very busy and labels within some of figures become very small (e.g. Fig. 3C). I suggest to subdivide it into two figures, e.g. by splitting of model (Fig. 3e).

Reviewer #2 (Remarks to the Author):

Singh et al use single-molecule FRET analysis to probe real-time interactions between RNA-programmed Cas9 and its DNA targets. The key finding is that the dissociation rate of Cas9 from DNA is highly dependent on mismatches between the guide RNA and the DNA sequence that are proximal to the PAM sequence. Mismatches distal to the PAM sequence having little to no effect on Cas9 binding to DNA. The authors also detect two different bound FRET states, which may represent distinct steps in target searching and/or proof reading.

Although, the concept of PAM, seed sequence and off-rate controlled binding have been presented before this study provides novel insight into the effects on mismatches between the target and the guide RNA, at the single-molecule level (at a resolution not before seen). Also novel is the

Detection of two bound FRET states, which provides evidence for a two-step mechanism of Cas9 DNA binding - the first step being PAM surveillance and the second involving RNA-DNA heteroduplex formation upon PAM recognition (which is also supported by recent structural data).

The experiments were expertly performed and interpreted. The data is of high quality and supports the conclusions presented. I recommend publication of this work as is.

We appreciated the constructive criticisms of the reviewers and have addressed their concerns by adding more text, explanations, references and figures. We hope that the revised manuscript will comply with the referees' remarks.

Point by point response, Reviewer #1:

- 1) The manuscript cites very few references despite rapidly growing number of studies that investigate Cas9 binding. As already stated above authors should compare in much more detail their achievements with respect to previous studies in CRISPR-field and to what extent the authors go beyond. The manuscript will certainly benefit from more comprehensive comparison that could also include R-loop formation in other CRISPR systems.**

The introduction of the article has been reworked with the addition of a new paragraph that references a majority of previous studies that have been conducted to investigate off-target effects of Cas9-RNA and related CRISPR-Cas design tools. Also included is a more detailed comparison with a single-molecule¹ study that investigated the Cas9-RNA induced RNA-DNA heteroduplex formation via magnetic tweezers, and found that 11 PAM-proximal matches are sufficient for stable RNA-DNA heteroduplex formation with StCas9 (the Cas9 ortholog from *Streptococcus thermophilus*). In our current study using SpCas9 (from *Streptococcus pyogenes*, simply referred to as Cas9), we found 9-10 PAM-proximal matches to be sufficient for ultra-stable Cas9-RNA binding. The stability of RNA guided CRISPR enzymes

and DNA targets depends on energetic contributions of the RNA-DNA heteroduplex and interactions between the DNA target and amino-acid residues of the CRISPR-Cas enzymes, which may explain subtle mechanistic differences between the various Cas9 orthologs.

- 2) **The authors report second bound state. How clearly defined is that state? It appears to me that authors only allow 3 states when obtaining their idealized FRET trajectories (e.g. Suppl. Fig 4b). Could it in reality be that second state actually consists of more states? Second state is very broadly distributed in transition density plots.**

The reviewer is correct in suggesting that the second bound state may consist of more than one state because the Cas9-RNA in complex with DNA during target search by definition would have multiple positions on the DNA, as exemplified by the broader distribution that the reviewer noted. The reason why we chose three states instead of four or more was because HMM analysis with four or more states did not give us additional discrete states (see new Supplementary Figure 6). By restricting the analysis to three states, we could group multiple states within the sampling mode into a single state, allowing us to study the impact of various mismatches systematically.

Supplementary Figure 6. Transition density plots for 9-20_{mm}, 5-20_{mm} and 1-2_{mm} DNA targets from two different inputs for hidden Markov modeling.

3 hidden Markov states were sufficient to capture the different FRET states of Cas9 targeting as any additional input state for hidden Markov modeling did not, evidently, result in any new discrete FRET state.

- 3) Please state more clearly what interpretation of precipitously decreased life time for PAM-proximal mismatches is. I suspect that in these cases R-loops that reach PAM-distal end are not forming at all, such that only target scanning occurs? Or do authors suspect destabilized R-loops due to presence of the PAM proximal mismatches?

This is a good point. In response, we have modified a sentence in the main text as following.

“The amplitude-weighted lifetime of the bound state (τ_{avg}) decreased precipitously even with just 2 bp PAM-proximal mismatches, likely because the R-loop failed to extend beyond the mismatches.”

- 4) **Figure 3 is very busy and labels within some of figures become very small (e.g. Fig. 3C). I suggest to subdivide it into two figures, e.g. by splitting of model (Fig. 3e).**

We thank the reviewer for this helpful suggestion. Figure 3 has now been divided into two separate main text figures, i.e. **Figure 3** with all the lifetime data and **Figure 4** with the kinetic model of Cas9-RNA targeting.

Point by point response, Reviewer #2:

- 5) **The experiments were expertly performed and interpreted. The data is of high quality and supports the conclusions presented. I recommend publication of this work as is.**

We are pleased with the reviewer's positive feedback.

References

1. Szczelkun, M.D. et al. Direct observation of R-loop formation by single RNA-guided Cas9 and Cascade effector complexes. Proceedings of the National Academy of Sciences of the United States of America **111**, 9798-9803 (2014).

2. Slaymaker, I.M. et al. Rationally engineered Cas9 nucleases with improved specificity. *Science* **351**, 84-8 (2016).
3. Kleinstiver, B.P. et al. High-fidelity CRISPR-Cas9 nucleases with no detectable genome-wide off-target effects. *Nature* **529**, 490-5 (2016).
4. Anders, C., Niewoehner, O., Duerst, A. & Jinek, M. Structural basis of PAM-dependent target DNA recognition by the Cas9 endonuclease. *Nature* **513**, 569-573 (2014).

REVIEWERS' COMMENTS:

Reviewer #1 (Remarks to the Author):

The authors have sufficiently addressed my previous concerns.